

# A conjecture on the minimal size of bound states

**Ben Freivogel[1⋆], Thomas Gasenzer[2,3†], Arthur Hebecker[3‡] and Sascha Leonhardt[3¶]**

**1** Institute for Theoretical Physics, University of Amsterdam,
Science Park 904, 1090 GL Amsterdam, Netherlands
**2** Kirchhoff-Institute for Physics, Heidelberg University,
Im Neuenheimer Feld 227, 69120 Heidelberg, Germany
**3** Institute for Theoretical Physics, Heidelberg University,
Philosophenweg 19, 69120 Heidelberg, Germany

⋆ benfreivogel@gmail.com
† t.gasenzer@thphys.uni-heidelberg.de
‡ a.hebecker@thphys.uni-heidelberg.de
¶ s.leonhardt@thphys.uni-heidelberg.de

## Abstract

We conjecture that, in a renormalizable effective quantum field theory where the heaviest stable particle has mass $m$, there are no bound states with radius below $1/m$ (Bound State Conjecture). We are motivated by the (scalar) Weak Gravity Conjecture, which can be read as a statement forbidding certain bound states. As we discuss, versions for uncharged particles and their generalizations have shortcomings. This leads us to the suggestion that one should only constrain rather than exclude bound objects. In the gravitational case, the resulting conjecture takes the sharp form of forbidding the adiabatic construction of black holes smaller than $1/m$. But this minimal bound-state radius remains non-trivial as $M_\mathrm{P} \to \infty$, leading us to suspect a feature of QFT rather than a quantum gravity constraint. We find support in a number of examples which we analyze at a parametric level.



# 1  Introduction

It is often useful to understand which phenomena or constructions can *not* arise within a given theoretical framework. Prominent examples are the CPT theorem in quantum field theory or the uniqueness of Einstein gravity as an interacting theory of spin-2 fields. Under the name 'Swampland program' [1–3] this line of research has recently gained momentum in string theory. One of its central claims is the weak gravity conjecture (WGC) [3][1] which relates gravitational and gauge forces. In this letter, our attempts to generalize this conjecture to all forces will lead us in an unexpected direction, making claims about the non-existence of certain bound states in quantum field theory (with little or no relation to gravity).

One may formulate the WGC by saying that, for any abelian gauge force, equal-charge particles are more strongly repelled by the gauge force than they are attracted by gravity.[2] By definition, this excludes a gravitational bound state of two or more such particles. Interestingly, it also implies that charged black holes, even extremal ones, can kinematically always decay. One may hence hope that the conjecture is actually more general, forbidding (under certain assumptions) the existence of stable bound states.

Indeed, it has been tried to extend the WGC along these lines to other forces, specifically to interactions mediated by a light scalar [19] (see also [20–28]). This is still fairly straightforward in the context of states charged under a gauge force, where the scalar force is merely an addition. One can then demand that two such particles are more strongly repelled by the gauge force than they are attracted by gravity together with the scalar force (which is always attractive).

If one starts talking about uncharged states, there is a problem: In this case, practically stable[3] bound states, such as planets, clearly exist. Hence, we will try not to forbid bound

---

[1] See [4–17] for some of the more recent work and [18] for an extensive review.

[2] In fact, this is oversimplified - one can only try to claim that a particle species exists for which the above statement holds.

[3] Stability relies on the conserved baryon number which, as an accidental global symmetry, should be weakly violated. By 'practically stable' we refer to the extremely large proton lifetime $\gtrsim 10^{33}$ y. We are after a practically useful generalization of the WGC and are therefore not satisfied with the possible and fairly obvious statement that all bound states not protected by gauge invariance must decay.

states in general but to constrain them. A benchmark case of a bound state is a boson star[4]. Specifically, consider a collection of $N$ free complex scalars, with the number $N$ protected by a global $U(1)$. Such stars become smaller as $N$ grows and collapse to black holes at a critical (minimal) radius $R \sim 1/m$, with $m$ the mass of the elementary bosons.

Based on this, a possible conjecture about forces is the following: They should not allow stable bound states with a radius that is smaller than what gravity can achieve. In other words, attractive forces can not lead to arbitrarily compact stars that can then collapse to arbitrarily small black holes. As a result, adiabatically produced black holes have a minimal size. An alternative formulation reads: In a quantum field theory where the heaviest elementary state has mass $m$, no bound state with radius below the scale $1/m$ set by the Compton wavelength exists. Thus, gravity may be dropped, the connection to the standard notion of the Swampland and the scalar WGC becomes remote, but we may have an interesting statement about quantum field theories. It appears plausible given its similarity to the uncertainty relation. Yet, it is still non-trivial since, in principle, a strong external force can of course confine a state on an arbitrarily short length scale. The claim is that in a consistent, fully dynamical, power-counting renormalizable field theory this can not happen.

If such a bound were proven to be true, a very interesting conclusion about the interplay of IR and UV physics in QFT may follow: There is no efficient, adiabatic, IR method of generating UV excitations. That is, we cannot adiabatically collect particles in the IR theory to gather enough energy to create particles of a UV completion of that theory. For this, one would have to collect the IR particles with mass $m$ in a spatial region of size $\ll 1/m$. The smallness is necessary to ensure a non-negligible transition rate of our collection of many light particles to few heavy particles of mass $M \gg m$. Thus, it may be possible to read our conjecture as forbidding such an 'IR-to-UV transformer'.

We focus on the case of $3 + 1$ dimensions, commenting briefly on the extension to other dimensions in the discussion section.

The rest of the paper is organized as follows. In Sect. 2, we briefly review ideas on how to constrain scalar interactions in the spirit of the swampland. We point out a possible counterexample to a recent conjecture and argue for a slightly different path towards promising constraints. The Bound State Conjecture is motivated and quantified in Sect. 3. As explained above, we base our discussion on the minimal size of a black hole built purely through gravitational attraction. We then formulate an inequality which bound states with general interaction should satisfy such that smaller black holes can not be constructed. Finally, we decouple gravity and arrive at a formulation of the Bound State Conjecture for general QFTs. Evidence for the conjecture is collected in Sect. 4. For this we consider both non-gravitational theories (a model with scalar-modulus coupling and $\phi^4$-theory) and gravitational theories (boson stars, where the scalar interaction adds to the gravitational attraction). We discuss our results and collect open questions in Sect. 5.

## 2 Constraining scalar interactions

### 2.1 From the WGC with scalars to scalar WGCs and beyond

Consider BPS states in an $\mathcal{N} = 2$ supergravity theory with vector multiplets (see e.g. [46]). Such states are (in general electrically and magnetically) charged under the vectors. They are also coupled to the vector-multiplet scalars or moduli $\chi_i$. This latter coupling can be understood on the basis of the moduli dependence of the mass squared, $m_\phi^2 = m_\phi^2(\chi_i)$, of the BPS state $\phi$. The effective Lagrangian term $m_\phi^2 \phi^2/2$ then induces the trilinear coupling

---

[4] For a review on boson stars (updated in 2017) see [29]. A selection of recent work can be found in [30–45].

$\mu_i m_\phi \chi_i \phi^2$ with $\mu_i \equiv \partial_i m_\phi$ [19]. In this language, the BPS relation of [46] takes the form

$$\text{mass}^2 + \sum (\text{scalar interaction couplings})^2 = \sum \text{charges}^2. \tag{1}$$

Here the 'mass' is to be taken in Planck units and the 'scalar interaction couplings' are the dimensionless quantities $\mu_i$. Given that the scalar mediator induces an attractive Coulomb force, this relation can be interpreted as force neutrality between (asymptotically separated) BPS particles. Repulsive gauge interaction and attractive gravitational plus scalar interaction cancel exactly.

A non-supersymmetric generalization of the above was proposed in [19]:

$$\text{mass}^2 + \sum (\text{scalar interaction couplings})^2 \leq \sum \text{charges}^2. \tag{2}$$

This is the 'Weak Gravity Conjecture with Scalar Fields'. By the force argument above, imposing such an inequality on charged particles forbids them to form stable bound states.[5] In [20] it was shown that, in a simple toy model, the WGC in 5d is equivalent to this inequality after compactification to 4d.

Next, one may wonder whether a bound of the form (2) exists for uncharged particles. The obvious candidate is

$$\text{mass}^2 + \sum (\text{scalar interaction couplings})^2 \leq 0. \tag{3}$$

This simply says that stable, uncharged, massive particles are forbidden.[6] Such a statement makes sense since long-lived particles are usually protected by global symmetries. But the latter are at best approximate, making the lifetime of the corresponding particles always finite. However, in practice global symmetries may be of excellent quality, as in the case of baryon number and the corresponding proton lifetime of about $10^{24} \times$ (Age of the Universe). Some uncharged particles may hence be 'practically stable' (we will refer to them as stable for simplicity), making a conjecture of the type of (3) somewhat boring.

In [19] it is speculated that the correct (and less obvious) ansatz for a bound on uncharged states is the 'Scalar Weak Gravity Conjecture' (SWGC)

$$\text{mass}^2 \leq \sum (\text{scalar interaction couplings})^2. \tag{4}$$

This can again be motivated by $\mathcal{N} = 2$ models, where a specific subclass of BPS states satisfies (4) with an equal sign. The obvious interpretation of the inequality (4) relates nicely to other versions of the WGC: Gravity is always the weakest force and is thus, in particular, also weaker than scalar attraction. Since both the left- and right-hand side characterize attractive forces, there is no interpretation in terms of a net repulsive force or a bound state argument. The conjecture may also appear problematic for another reason: In the effective theory below the mass scale of the exchange scalar (which will be non-zero without $\mathcal{N} = 2$ SUSY), the conjecture cannot even be formulated. Thus, this conjecture is on weaker footing.

In [24], it is suggested to apply (4) to scalar particles with self-interactions and to interpret it as an inequality for derivatives of the potential. In this case, the mass squared corresponds to $V''$ and the trilinear coupling to $V'''/\sqrt{V''}$. Thus, an inequality of the type $(V'')^2/M_P^2 \leq 2(V''')^2$

---

[5] To be precise, one only demands that *some* of the charged particles must satisfy the inequality. As a result, a finite number of bound states may still be allowed.

Furthermore, one needs the scalar fields to be exactly massless for this interpretation. However, without $\mathcal{N} = 2$ SUSY such scalars presumably do not exist. It is then not completely clear how to formulate a fundamental principle supporting a 'WGC with scalars' since the scalar force effect disappears at asymptotically large distances. We do not discuss this further since our suggested scalar conjecture will in any case be rather different.

[6] Particles charged under a *discrete* gauge symmetry can certainly be stable and should be allowed together with $U(1)$-charged particles.

results. Since a quartic interaction with negative sign also represents an attractive force, the authors supplement the above by a corresponding 4th-order derivative term,

$$(V'')^2/M_{\mathrm{P}}^2 \;\leq\; 2(V''')^2 - V''''V''. \tag{5}$$

This is proposed as the 'strong Scalar Weak Gravity Conjecture' (sSWGC). Recently, a 'generalized Scalar Weak Gravity Conjecture' based on the sSWGC has been put forward and applied to early universe dynamics [28].

## 2.2 Possible counterexamples

It is easily seen that the sSWGC leads to very strong constraints in some cases. For example, for the potential

$$V(\phi) = \frac{1}{2}m^2\phi^2 + \lambda\phi^4, \tag{6}$$

with positive mass squared, it implies that

$$m^2/M_{\mathrm{P}}^2 \leq -\lambda, \tag{7}$$

if one analyzes the point $\phi = 0$ [7]. This would allow only for attractive interactions ($\lambda < 0$) between massive ($m^2 > 0$) scalars described by such a potential. However, we expect this potential to be in the landscape for any sign of the interaction, as it is the most general power-counting renormalizable potential for a scalar with a $\mathbb{Z}_2$ symmetry $\phi \to -\phi$.

Indeed, and this is our first new technical point, a dilute gas of scalar atoms (e.g. helium-4) provides such a counterexample: Such light atoms are characterized by mass and radius on the order of

$$m \approx 10^3\,\mathrm{MeV}, \qquad R \approx 1\,\mathrm{nm} \approx 10^3/\mathrm{MeV}, \tag{8}$$

respectively. Below the energy scale $1/R$, they behave as point-like particles with a 2-to-2 repulsive $\delta$-function interaction. Such a system can be described by an effective quantum field theory based on the Lagrangian $\mathcal{L}(\phi, \dot{\phi}) = (\partial \phi)^2/2 - V(\phi)$, where the potential is as in (6), with positive coupling $\lambda > 0$. Here a real field is sufficient if only particles (and no antiparticles) are present and the energy scale $1/R$ is too low for pair production.

We start by a first estimate of the violation. The scattering cross section of this QFT system is then $\sigma_{2\to2} \sim \lambda^2/m^2$, to be compared with the hard-sphere value $\sigma_{\mathrm{geom}} \sim R^2$. Identifying the cross sections $\sigma_{2\to2}$ and $\sigma_{\mathrm{geom}}$ gives $\lambda \sim Rm \sim 10^6$. In the regime of small $\phi$ the conjecture demands (7). The explicit violation then reads

$$10^{-36} \nleq -10^6. \tag{9}$$

It arises simply because at low density and energy, corresponding to $\phi \approx 0$, we have $\lambda > 0$.

In spite of the large values of the coupling constant $\lambda$, we argue that at low energies and density (6) is a perfectly fine quantum field theory which is realized in nature (e.g., by helium-4 atoms, or, as extensively demonstrated in experiments, by dilute alkali gases which have scattering lengths of similar size) and hence provides a counterexample to (5). To be more precise about the density requirement, we note that a gas with mean particle distance $l$ has an energy density $m/l^3$. If the state is sufficiently homogeneous and coherent to allow for a classical field description, this density can be identified with $m^2\phi^2$. Thus, the field value is

---

[7] For non-vanishing but small field values $\phi^2 \ll m^2/|\lambda|$, we have $\frac{m^2}{M_{\mathrm{P}}^2} \leq -\lambda\left(1 + \mathcal{O}\left(\frac{\phi^2}{m^2/|\lambda|}\right)\right)$.

set by $\phi^2 \sim 1/ml^3$. We now see that perturbativity (in the sense that $m^2\phi^2$ dominates over $\lambda\phi^4$) is guaranteed when the gas is dilute (in the sense that $l \gg R$) [8]:

$$\frac{\lambda\phi^4}{m^2\phi^2} \sim \frac{\lambda/m^2 l^6}{1/ml^3} \sim \frac{\lambda}{(ml)^3} \sim \frac{(R/l)^3}{(mR)^2} \ll (R/l)^3. \tag{10}$$

In other words, diluteness (and small energies) implies $\phi^2 \ll m^2/\lambda$, which gives rise to the approximation used in calculating (7).

Alternatively, and maybe more appropriately in this atomic-physics context, one can approximate the corresponding equation of motion of this $\lambda\phi^4$-theory by the non-relativistic Gross-Pitaevskii (GP) equation [47, 48]

$$i\partial_t \varphi = \left(-\frac{\nabla^2}{2m} + g\,|\varphi|^2\right)\varphi. \tag{11}$$

Here $\varphi \simeq (m\phi + i\pi)/\sqrt{2m}$ (with $\pi \equiv \dot{\phi}$) is the non-relativistic, normalized complex scalar field and $g = \lambda/8m^2$ is the GP coupling. The 'radius' of the interacting atoms in a quantum mechanical treatment corresponds to the ($s$-wave) scattering length $a$ entering the low-energy field theory through $a = mg/(4\pi) = \lambda/(32\pi m)$. $a$ is defined via the phase shift between incoming and outgoing wave in the low-energy limit [49]. Applied to hard spheres of radius $R$, this definition indeed gives $a \equiv R$. With this more precise description, one finds, for helium-4 atoms in the ground state, that $a \approx 8\,\text{nm}$ [50, 51]. The relativistic coupling is then positive and thus repulsive, with $\lambda \approx 2 \cdot 10^9$ (using $m_{(^4\text{He})} \approx 4\,\text{GeV}$).

Let us comment on a possible modification of the statement of the sSWGC (5). In the spirit of the original SWGC (4) one might expect the inequality to be valid only for massless exchange particles. For a self-interacting theory of massive particles, this would correspond to requiring that the inequality is satisfied only at energy scales far beyond the particle's mass. In this way, the non-relativistic atomic gas described above would not be subject to the modified conjecture.

There still seem to be theories that are expected to be in the landscape but are in tension with even this modified sSWGC. Take again the self-interacting theory (6). As already alluded to in [24], one can check that near the vacuum configuration, $\phi^2 \ll \left|m^2/\lambda\right|$, the sSWGC implies that the product $\lambda m^2$ is negative. One expects though that a generic supersymmetric theory with soft mass terms and quartic interactions from D-terms can have both positive $m^2$ and $\lambda$ since the parameters can arise from independent sectors.

## 3 Bound State Conjecture

### 3.1 An alternative approach to constraining scalar interactions

As we have tried to explain, we perceive the idea of constraining attractive scalar forces as very promising but are not fully convinced by the corresponding conjectures proposed until now. More concretely, there exists a straightforward logic taking us from the WGC, via the BPS condition (1), to (2). However, the next steps are less clear: While (3) could be called trivial, (4) and (5) appear problematic. Especially the crucial sign-flip of the scalar force between (2) and (4) may lack motivation.

Thus, we return to the WGC with scalars, as quantified by (2), and generalize it in an entirely different way. We propose that, in the absence of charges, the conjecture still constrains the strength of attractive forces and hence bound states, but in a less trivial way than

---

[8] This is true for either sign of $\lambda$. The equation can be read with $\lambda$ replaced by $|\lambda|$. Instead of a hard-shell radius $R$ one should insert the absolute value of the (possibly negative) s-wave scattering length $a$, see below.

(3). Namely, we accept that uncharged states *can* be bound by gravity (as it happens with the practically stable neutral atoms of the real world). However, we conjecture that any additional attractive force *can not* enhance this binding parametrically. A way to quantify this is through the size of the smallest black hole which can be adiabatically built from a given particle species. Indeed, stronger gravity (smaller $M_\mathrm{P}$ or larger particle mass) allows for building smaller black holes. Attractive QFT forces can hence be constrained by imposing a lower bound on their radius. We phrase this as the **Bound State Conjecture:**

> *In a theory where the heaviest stable particle has mass m, it is impossible to construct adiabatically a black hole that is parametrically smaller than the black hole that can be built from free scalars of the same mass m.*

Let us argue more carefully why this constrains scalar forces: Consider a theory with just gravity and a free scalar field. Furthermore, consider a cloud of gravitationally bound scalar particles (a boson star) and a quasi-stationary process in which particles are added. In this process, the cloud becomes smaller and denser, eventually collapsing to a black hole. A very weak, additional attractive force does not change this picture qualitatively but makes the star at each stage even smaller, eventually leading to a smaller black hole. Our claim is that this road to smaller bound objects is severely limited: The purely gravitational case cannot be beaten parametrically. In fact, we would like to claim this in full generality, allowing for multiple particle species, gauge and scalar forces, quartic and any other interactions: In any given field theory coupled to gravity, no bound state parametrically smaller than the collapse size of a purely gravitational boson star made of the heaviest stable particle species [9] can be constructed. Due to gravity, there is, however, one exception: Following a black hole collapse Hawking evaporation sets in. So we can indeed create black holes smaller than $1/m$, but our conjecture is precisely about this *only* being possible due to gravity and Hawking evaporation (see also Sect. 3.2). Additional attractive forces within a QFT can not achieve this.

The possible bearing of the swampland idea on boson stars has recently been discussed in [52,53]. In particular, the swampland distance and de Sitter conjectures attempt to constrain scalar dynamics and are hence relevant to what type of boson stars may exist [53]. At the moment, we do not see a direct connection to our approach, but this may change with further research.

## 3.2 Quantifying the Bound State Conjecture

To make our conjecture more explicit, we need to know the mass of the smallest black hole that can be adiabatically constructed in a theory of free (up to gravitational interactions) bosons. It is simplest to use complex free bosons, where a global $U(1)$ symmetry ensures approximate particle number conservation (up to very small non-perturbative gravitational effects). In this setting, we want to determine the size of a cloud of $N$ gravitationally bound bosons of mass $m$. Assuming them to be confined in a sphere of radius $R$, we can estimate the energy (ignoring $\mathcal{O}(1)$-factors) by

$$E_\mathrm{tot} = E_\mathrm{grav} + E_\mathrm{loc} \sim -\frac{M^2}{M_\mathrm{P}^2 R} + N\frac{p^2}{m} \sim -\frac{M^2}{M_\mathrm{P}^2 R} + \frac{N}{mR^2}. \tag{12}$$

Here we have added naive parametric estimates of gravitational binding and localization energy, the latter encoding what is also known as 'quantum pressure'. We use the uncertainty-

---

[9] Here, a particle is any state with mass $m$ and radius $\lesssim 1/m$. By radius we mean the length scale above which no sub-structure can be resolved. Concretely, one may think of gravitons scattering off the state and determining above which energy scale the form factor becomes non-trivial.

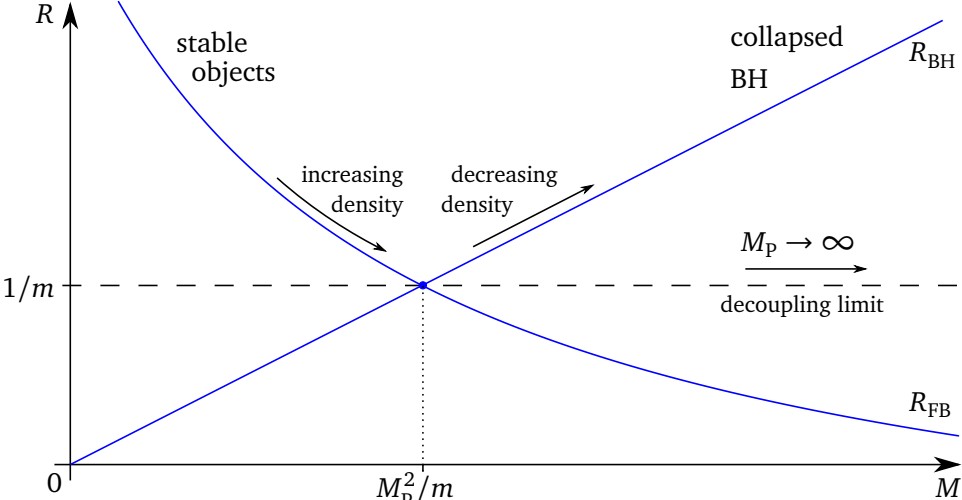

Figure 1: The mass-radius plot for the free boson star $R_{\mathrm{FB}}(M)$ and a black hole $R_{\mathrm{BH}}(M)$. As gravity is decoupled ($M_{\mathrm{P}} \to \infty$) the intersection point of the two curves moves on the dashed line.

principle value $p = p(R) \sim 1/R$.

Our expectation is that the equilibrium situation corresponds to roughly the minimum of (12) as a function of $R$. Using $N = M/m$, this 'free boson star radius' is found to be

$$R_{\mathrm{FB}}(M) \sim \frac{1}{M} \left( \frac{M_{\mathrm{P}}}{m} \right)^2, \tag{13}$$

in agreement with numerical results [54, 55]. Thus, as we keep adding particles and thereby increasing $N$ and $M$, the boson star becomes denser (cf. Fig. 1). At some point, it reaches the critical radius [10] of black-hole collapse $R_{\mathrm{FB}} \sim R_{\mathrm{BH}} = M/M_{\mathrm{P}}^2$. As a consequence, the density falls off again as even more mass is added, as is common for black holes. Of course, once we deal with a black hole, Hawking radiation also allows for moving downwards on the straight 'black hole' line in Fig. 1, as already stated in Sect. 3.1.

With this we can now quantify the Bound State Conjecture in the following way. A stable boson star built by a quasi-stationary process in any (in general interacting) theory gives rise to a curve $R(M)$. The intersection of $R_{\mathrm{BH}}(M)$ with $R(M)$ in a graph analogous to Fig. 1 defines a minimum size $R^{\mathrm{min}}$. At this radius black-hole collapse sets in. The conjecture demands that the resulting black hole is not parametrically smaller than the minimal size of a black hole built from free scalars of the same mass. This amounts to requiring

$$R^{\mathrm{min}} \gtrsim R_{\mathrm{FB}}^{\mathrm{min}} \sim \frac{1}{m}. \tag{14}$$

In other words, the intersection of $R(M)$ with $R_{\mathrm{BH}}(M)$ cannot lie parametrically below the dashed line in Fig. 1.

Finally, we should comment on the definition of the radius $R(M)$ of a boson star. In a field theory, this radius can of course only be a typical length scale since field profiles are infinitely extended. In numerical calculations this scale is usually taken to be $R_{99}$, the radius which contains 99% of the mass. Since we are working only at the parametric or $\mathcal{O}(1)$-level, an analogously defined radius $R_{50}$ is presumably more useful in our context.

---

[10] Inserting more correctly the Buchdahl limit or similar bounds results in further numerical $\mathcal{O}(1)$-factors only. We ignore this subtlety.

## 3.3   Limit of decoupled gravity: General Bound State Conjecture

It is clear from the constraint on the size of bound states in (14) that our conjecture also makes sense in the limit of decoupled gravity, $M_P \to \infty$, since the dashed line in Fig. 1 is independent of $M_P$. It then says that any scalar configuration of arbitrary mass cannot be more localized than $R(M) \sim 1/m$ parametrically.

With this, and postponing a discussion of non-scalar bound states to Sect. 4.4, we are now in a position to quantify the (generalized) Bound State Conjecture:

---

**Bound State Conjecture**: *The typical radius R of a stable bound state in a power-counting renormalizable effective field theory, valid below some scale Λ, is bounded from below by*

$$R \gtrsim \frac{1}{m}. \tag{15}$$

*Here, the scale m ($\ll$ Λ) is the mass of the heaviest stable particle.*

---

We consider only renormalizable theories, as we are looking to constrain infrared physics far below the cut-off scale Λ. Any higher-order term in an effective Lagrangian will be suppressed by this high scale. We will consider a non-renormalizable example in Sect. 4.2.4, which will provide an a posteriori justification for this restriction in our conjecture.

Let us note that our conjecture does not constrain cases where particles with mass $m$ combine into a state of smaller mass, $M < m$. In this case, if $R(M) \lesssim 1/m$, then the newly formed object is, by our definition, a particle rather than a bound state. Furthermore, if $M > m$ one might worry that very small bound state radii, $R(M) \lesssim 1/M \lesssim 1/m$, would be allowed since, once again, the bound state must be viewed as a particle. However, this situation is excluded by the assumptions of our conjecture, which state that the particle of mass $m$ is the heaviest stable particle of our theory.

We have to be careful with the notion of the 'heaviest stable particle' when going beyond perturbative and purely scalar theories. In a theory with only massless fundamental fields, like QCD (with massless quarks), states with finite radius and non-zero mass can certainly exist. A particularly simple example would be the lightest scalar glueball in pure Yang-Mills theory. Given some additional gauge or scalar force, such objects can form bound states of non-zero radius. So it would be wrong to demand that our conjecture holds with $m = 0$. Indeed, we were careful to define the term 'particle' as a state which appears point-like below its own mass scale. This is in contrast to bound states, which are extended objects. Then e.g. a glueball or hadron can be viewed as a stable particle [11] whose mass sets the scale of the conjecture.

The bound states our conjecture applies to can, in general, carry higher spin. Recently, constraints have been put forward on the existence of higher-spin, composite states of mass $M$ and size $R$ which appear point-like or elementary in the sense that $R \lesssim 1/M$ [56, 57]. To make contact with our conjecture, which is specifically about non-elementary bound states, a generalization of the causality arguments of [56, 57] would be needed. Note that our constraint is stronger as it excludes bound states of size $R \lesssim 1/m$, with $m$ the mass of the heaviest stable constituent particle.

It may be possible to formulate our conjecture as a field-theoretic realization of a resource theory [58, 59]. To explain this, let us for simplicity consider a model with light fundamental fields of mass $\leq m$ and one heavy field of mass $M$. We can now introduce an artificial cutoff Λ, with $m \ll Λ \ll M$, and focus on the effective theory below Λ. Our conjecture forbids the adiabatic construction of small bound states from particles with mass $\leq m$, which are the only ones available in the IR effective theory. Here 'small' means smaller than the inverse of the

---

[11] Note again that our use of the word 'stable' includes objects protected by an approximate global symmetry.

maximum mass, $1/m$. We expect that, in the full theory, this implies that we cannot efficiently create particles with mass $M$, point-like on length scales $\gtrsim 1/M$, from particles with mass $\le m$ by an adiabatic process. The reason is that the transition rate from an extended object to a heavy fundamental particle is exponentially suppressed, even if quantum numbers allow the process in principle. For example, even if baryon number would be strongly violated, a small crystal of mass $M_{\text{P}}$ would certainly have a negligible transition rate to a Planck-scale fundamental particle. That would change if one could confine enough light particles at the required small length scale $\ll 1/m$, but this is precisely what our conjecture forbids. In the language of resource theory, we may hence define the free states as those normally available to an IR experimenter, i.e. the particles with mass $\le m$. The free operations would be adiabatic processes with those particles. Within this formulation and under these conditions, our conjecture is that a heavy particle with mass $M$ is a resource which can not be created by the IR experimenter.

Consider, e.g., a free stable boson star with radius (13) and mass $M \ll M_{\text{P}}^2/m$ which, in mean-field approximation, is expected to be well described by a (semi-classical) coherent state of $N \sim M/m$ bosons in a single spatial field mode. Forcing the particles with mass $m$ to localize at a smaller length scale $R$, with $R_{\text{BH}}(M) \ll R \ll 1/m$ requires a deformation of the energy function which leads to a strong squeezing of the phase-space distribution of each particle and thus of the whole star. The external modification must in particular allow for anti-squeezing perpendicular to the squeezed spatial extent of the Wigner distribution. This is similar to what a larger value of $m$ would achieve in the energetic balance (12), except that the squeezed particles assume relativistic momenta. Given that the modification is achieved by some in general non-linear interactions, a proof of our conjecture could then include an answer to the question why squeezing into the regime of relativistic momenta is inhibited, based on the practical impossibility to create the entanglement associated with the squeezed many-particle state.

One may summarize this in the **Bound State Conjecture (Resource-Theory Version)**: In a power-counting renormalizable effective theory where free states are those with particles of mass $\le m$ and free operations are adiabatic processes involving these particles, heavy particles with mass $M \gg m$ represent a resource.

## 4 Evidence from examples

We now discuss three different classes of examples all of which, as it turns out, appear to respect the conjecture: Attractive scalar-modulus couplings, quartic self-interactions of a complex scalar, and the attractive interaction implicit in axionic potentials.

### 4.1 Non-gravitational scalar-modulus coupling

Consider a theory with scalar interaction of the type discussed in [19, 20],

$$V(\phi) = \frac{m^2}{2}|\phi|^2 + \frac{\widetilde{m}^2}{2}\chi^2 + \mu m|\phi|^2 \chi, \tag{16}$$

where $\phi$ is a $U(1)$-symmetric complex scalar field [12] and the real field $\chi$ is very light, $\widetilde{m} \ll m$. We will think of $\chi$ as a modulus, mediating a long-range, attractive force which is capable of binding $\phi$ particles. In this spirit, we will neglect $\widetilde{m}$ whenever it gives only subleading

---

[12] Note that we use the same symbol $\phi$ for both complex and real scalar fields throughout the paper. We will clearly state in each section whether we deal with a real or complex theory.

corrections.[13] Note that, instead of $m$ and the dimensionless coupling $\mu$, we may also use the two dimensionful parameters $m$ and $\gamma \equiv \mu m$ to characterize the theory.

We introduced $\phi$ as a complex field so that we can make use of the conserved global $U(1)$ charge $N$ (and accordingly particle number conservation). Using the standard stationary ansatz $\phi(t, \mathbf{x}) = \phi(\mathbf{x})e^{-i\omega t}$ with $\phi(\mathbf{x})$ real (cf. [60, 61]), the particle number is given by

$$N = i \int \mathrm{d}^3 x \, (\phi^\dagger \dot{\phi} - \dot{\phi}^\dagger \phi) \sim \omega \int \mathrm{d}^3 x \, \phi(\mathbf{x})^2. \tag{17}$$

### 4.1.1 Non-relativistic estimate

As already discussed in the case of a gravitationally bound boson star, bound states with small $N$ are large and become smaller as $N$ grows. We start our analysis in the regime where $m \gg 1/R \gg \tilde{m}$. This allows for a simple controlled calculation since we are on the one hand safely non-relativistic and can, on the other hand, neglect the finite range of the force inside the bound state. Indeed, the $\chi$-exchange gives rise to an effective attractive potential of the Yukawa type, $V(r) \sim (\mu^2/r)e^{-\tilde{m}r}$, for the $\phi$ particles. The effective binding energy inside a boson star of $N$ particles is therefore estimated as [14]

$$E_{\mathrm{scalar}} \sim -\frac{N^2 \mu^2}{R} e^{-\tilde{m}R} \approx -\frac{N^2 \mu^2}{R}. \tag{18}$$

In addition, we have the localization energy (cf. (12))

$$E_{\mathrm{loc}} \sim N \cdot \frac{p^2}{m} \sim \frac{N}{mR^2}. \tag{19}$$

Minimizing the total energy as we did in the free case (12), we find the radius of this bound state with scalar force to be

$$R_{\mathrm{SF}}(N) \sim \frac{1}{N\mu^2 m}. \tag{20}$$

Given that the configuration is assumed to be spherically symmetric and localized inside a radius $R$ as well as using the non-relativistic frequency $\omega \sim m$,[15] we have according to the general result (17)

$$N \sim m \int_0^R \mathrm{d}r \, r^2 \, \phi(r)^2. \tag{21}$$

We expect that a $\phi(r)$-configuration sources a corresponding stationary profile $\chi(r)$. The latter is determined by the equation of motion

$$(\nabla^2 - \tilde{m}^2)\chi(r) = \mu m \, \phi(r)^2. \tag{22}$$

The Green's function for the operator on the l.h.s. is

$$G(r) = -\frac{e^{-\tilde{m}r}}{4\pi r}, \tag{23}$$

---

[13] We did not set $\tilde{m} = 0$ since, without $\mathcal{N} = 2$ SUSY, this is probably inconsistent. Moreover, it would be inconvenient to work in a model where only the boundary condition at spatial infinity sets the vacuum value of $\chi$.

[14] One can replace $\phi$ in $\mu m |\phi|^2 \chi$ with the non-relativistic field $\varphi \simeq (m\phi + i\dot{\phi})/\sqrt{2m} \sim \sqrt{2m}\phi$. Here, in the last equality we used the exponential ansatz with $\omega \sim m$. Now the interaction term takes the form $\mu |\varphi|^2 \chi$, explaining that only $\mu^2$ and no $m^2$ appears in (18).

[15] The frequency $\omega \sim m$ gives a solution to the equation of motion $(\nabla^2 - m^2 - \mu m \chi)\phi(r) = -\omega^2 \phi(r)$ for large $R$ (small gradients) and negligible interaction. The latter assumption can be checked a posteriori given the magnitude of back-reaction on the field $\chi(r)$.

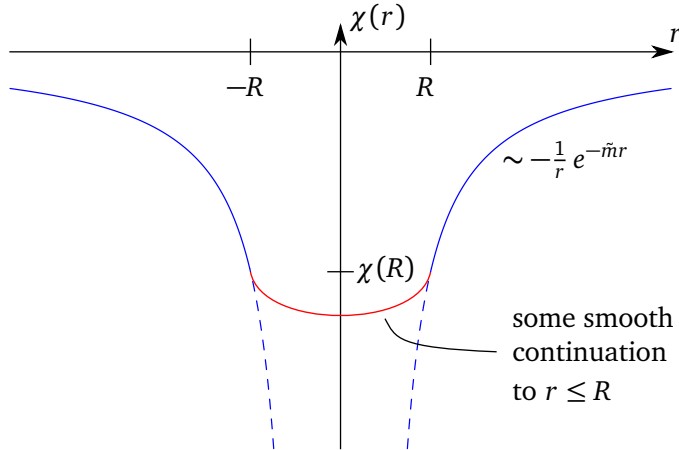

Figure 2: The $\chi(r)$-profile induced by the localized source $|\phi(r)|^2$. For better illustration, we extend the profile to $r < 0$ using spherical symmetry.

leading to

$$\chi(r) \sim -\mu m \int_0^R \mathrm{d}r'\, r'^2\, \frac{e^{-\tilde{m}(r-r')}}{r-r'} \phi(r')^2. \tag{24}$$

We assume that the source term is significant only inside a region of size $R$. Hence, focusing on the $\chi$ profile outside the star, we can use $r \gg R \geq r'$. This implies

$$\int_0^R \mathrm{d}r'\, r'^2 \frac{e^{-\tilde{m}(r-r')}}{r-r'} \phi(r')^2 \approx \frac{e^{-\tilde{m}r}}{r} \int_0^R \mathrm{d}r'\, r'^2 \phi(r')^2 \sim \frac{e^{-\tilde{m}r}}{r} \frac{N}{m}, \tag{25}$$

where we used (21) in the last step. This gives

$$\chi(r) \sim -\frac{N\mu}{r} e^{-\tilde{m}r}. \tag{26}$$

From this we can not determine the solution inside the star as this needs precise knowledge of the $\phi(r)$-profile. However, it is reasonable to assume that it is not parametrically larger than (26) at the boundary $r = R$ (cf. Fig. 2). Thus,

$$\chi(r \leq R) \sim \chi(R) \sim -\frac{N\mu}{R} e^{-\tilde{m}R} \approx -\frac{N\mu}{R}. \tag{27}$$

With this, we may reverse the logic and consider the effective potential for $\phi$, induced in part by the $\chi$ profile which we just determined:

$$V(\phi) \sim m^2 |\phi(t,r)|^2 \left(1 + \frac{\mu}{m}\chi(r)\right) \sim m^2 |\phi(t,r)|^2 \left(1 - \frac{N\mu^2}{mR}\right) \sim |\phi(t,r)|^2 \left(m^2 - \frac{1}{R^2}\right). \tag{28}$$

In the last two expressions, we assumed $r \lesssim R$ and made use of (20). This reveals a potential tachyonic instability. We see that the potential is only safely stable as long as

$$R_{\mathrm{SF}} \gtrsim \frac{1}{m}. \tag{29}$$

That is, for large $R$ the calculation can be trusted and stable bound states exist. By contrast, for $R \sim 1/m$ or smaller, the effective potential for $\phi$ in (28) tends to become tachyonic. This appears to support our conjecture. However, at the same time our non-relativistic approach breaks down. In the next subsection, we address this issue.

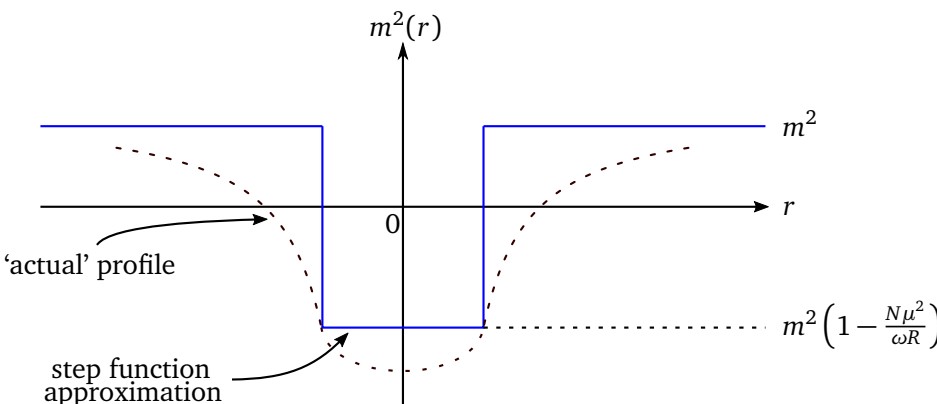

Figure 3: The effective mass squared for $\phi$ induced by the back-reaction on $\chi(r)$. We again extend the profile to $r < 0$ using spherical symmetry.

### 4.1.2 Analysis of the instability

There are two caveats to the above conclusion. The first one is that our non-relativistic calculation breaks down close to the regime of interest $R_{\text{SF}} \sim 1/m$. To remedy this non-relativistic assumption, we will repeat the calculation without specifying the oscillator frequency $\omega$ of the mode described by the mean field $\phi(t, r)$. The second is that, even if we continued to trust our calculation, the potential tachyonic instability found above does not necessarily lead to unstable bound states: While we found that *locally* the effective potential $m^2(r)\phi(r)^2 \sim m^2[1 + \mu\chi(r)/m]\phi(r)^2$ becomes tachyonic, this does not automatically lead to tachyonic modes. The oscillator frequency squared $\omega^2$ receives also a *positive* contribution from the field gradient, which can potentially maintain stability, i.e. $\omega^2 > 0$. To find the conditions for instability, $\omega^2 < 0$, we analyze more precisely the $\phi$-solutions in the $\chi$-background created in the stable regime.

To start, we again assume some localized $\phi$-configuration that sources a $\chi$-profile. The calculation is the same as before, now however, we insert (21) for general frequency $\omega$ to find

$$\chi(r) \sim -\frac{N\mu m}{\omega r} e^{-\widetilde{m}r}. \tag{30}$$

As before, we take the $\chi$-profile for $r < R$ to be some smooth continuation of the calculable exponential profile at $r > R$. In particular, $\chi(r < R) \sim -\frac{N\mu m}{\omega R}$, such that the effective $\phi$-potential for $r \lesssim R$ is

$$V(\phi) = \frac{m^2}{2}|\phi|^2 + \mu m \chi(r)|\phi|^2 \sim m^2\left(1 - \frac{N\mu^2}{\omega R}\right)|\phi|^2. \tag{31}$$

We will simplify this even further by using a step function approximation for the effective mass squared $m^2(r) = m^2[1 + \mu\chi(r)/m]$, see Fig. 3.

To find the radius that minimizes the energy, we equate the absolute values of the localization energy

$$E_{\text{loc}} \sim \int \mathrm{d}^3x \, |\nabla\phi|^2 \sim \frac{1}{R^2}\int \mathrm{d}^3x \, \phi^2 \sim \frac{N}{\omega R^2} \tag{32}$$

and the binding energy

$$E_{\text{scalar}} \sim \int \mathrm{d}^3x \, \mu m \chi \, |\phi|^2 \sim \mu m \, \chi(r \lesssim R) \int \mathrm{d}^3x \, \phi^2 \sim -\frac{N^2\mu^2 m^2}{\omega^2 R}. \tag{33}$$

The resulting radius of the scalar-force bound state scales as

$$R_{\mathrm{SF}}(N) \sim \frac{\omega}{N\mu^2 m^2}. \tag{34}$$

As we have $\omega = \omega(R)$ and minimize the energy, we have to assume that the derivatives $\omega'(R)$ are sufficiently well-behaved to arrive at (34). In principle, we should also have taken into account

$$E_{\mathrm{mass}} \sim \int \mathrm{d}^3 x \, m^2 \, |\phi|^2 \sim \frac{m^2 N}{\omega}. \tag{35}$$

This term however, is just a constant offset in the non-relativistic regime, where $\omega \sim m$, and becomes negligible compared to $E_{\mathrm{loc}}$ in the relativistic regime, once $R \lesssim 1/m$.

Finally, we check the results obtained so far for consistency: We insert our calculated $\chi$-profile, which depends on $\omega$, into the equation of motion for $\phi$ and check whether this indeed produces a stable lowest-energy localized $\phi$-mode with oscillator frequency $\omega$. The stationary ansatz $\phi(t,r) = e^{-i\omega t}\phi(r)$ together with the back-reacted effective mass squared lead to the equation of motion

$$\left[\omega^2 + \nabla^2 - m^2(r)\right]\phi(r) = 0. \tag{36}$$

The instability expected in the previous subsection is realized if we find negative eigenvalue solutions, $\omega^2 < 0$. Such solutions would signal tachyonic modes which are not stationary and do not give rise to stable bound states.

At this point, we should highlight that we are actually doing a quantum rather than a purely classical calculation: Since we are dealing with bosons, we expect all the particles to populate the lowest (quantum) mode. The coherent state of bosons in the ground state has the interpretation of a classical field configuration described by the underlying wave function. This wave function, the lowest lying solution of the above classical equation of motion, must have positive 'energy' $\omega^2$ for this standard interpretation to apply. But if $\omega^2$ is negative, our bosons do not populate a positive-frequency-squared but rather a negative-frequency-squared oscillator. This is the quantum interpretation of the possible instability.

We notice that (36) represents a 3d Schrödinger-type equation when identifying

$$E \equiv \omega^2 - m^2 + \frac{\alpha}{R^2}, \qquad V(\mathbf{x}) = V(r) \equiv \frac{\alpha}{R^2}\theta(r - \beta R). \tag{37}$$

Here, we explicitly reintroduced unknown $\mathcal{O}(1)$-coefficients $\alpha$ and $\beta$ appearing in the $\chi$-profile, $V_0 = \alpha/R^2$, and the radius, $R_{\mathrm{actual}} = \beta R$. We can reduce this to a one-dimensional problem by introducing $u(r) = r\phi(r)$, which fulfills the Schrödinger equation,

$$\left[E + \partial_r^2 - V(r)\right]u(r) = 0, \tag{38}$$

on $\mathbb{R}^+$ and is subject to the boundary condition $u(0) = 0$. The potential $V(r)$ vanishes for $r \leq \beta R$ and has a finite height $V_0 = \alpha/R^2$ otherwise. The energies corresponding to bound-state solutions obey $0 \leq E \leq V_0$ or equivalently $m^2 - \frac{\alpha}{R^2} \leq \omega^2 \leq m^2$.

An elementary textbook-style analysis of (38) shows that the energy eigenvalues fulfill

$$\sqrt{m^2 - \omega^2} = -\sqrt{\omega^2 - m^2 + \alpha/R^2}\cot\left((\beta R)\sqrt{2(\omega^2 - m^2 + \alpha/R^2)}\right). \tag{39}$$

A necessary condition for a solution is that the zero of the l.h. side of (39) is larger than the first zero of the r.h. expression. That is

$$m^2 \geq m^2 + \left(\frac{\pi^2}{8\beta^2} - \alpha\right)\frac{1}{R^2}. \tag{40}$$

Depending on the unknown coefficients this may or may not hold. Our discussion in Sect. 4.1.1 showed that solutions exist in the non-relativistic regime $R \gg 1/m$. When moving into the regime $R \sim 1/m$, the coefficients $\alpha$ and $\beta$ stay roughly the same and we expect that not all solutions disappear. A physical reason that a stable localized solution should prevail even for $R \ll 1/m$ is as follows: Let us bring a particle (adiabatically, i.e. without kinetic energy) close to the localized configuration, or rather the potential well created by it. The latter corresponds to an attractive force. If, say, $N$ particles have been bound in this way, there is no reason for the $(N+1)$st particle coming from infinity not to feel an attractive potential and be bound as well.

We parameterize the energy of the lowest mode by

$$E = \gamma V_0 = \gamma \frac{\alpha}{R^2}, \qquad \omega^2 = m^2 + (\gamma - 1)\frac{\alpha}{R^2}, \tag{41}$$

with some factor $0 < \gamma \leq 1$ which depends on $\alpha, \beta$ and is fixed by (39). The value $\gamma = 1$ corresponds to marginal binding, when the equality applies in (40).

We see from the 2nd equation in (41) that, in the non-relativistic regime $R \gg 1/m$, any value of $\gamma$ leads to a solution with $\omega^2 \sim m^2$. In the relativistic regime, $R \ll 1/m$, our analysis allows for two scenarios: On the one hand the solution can become tachyonic. This happens if $\gamma$ takes generic values $0 < \gamma < 1$. On the other hand, the value of $\gamma$ could always come out to be very close to unity, such that we continue to have a positive $\omega^2$ solution and the bound state remains stable. In fact, the level of tuning required for this grows with decreasing $R$: To have a non-tachyonic solution, we need $0 \leq 1 - \gamma < m^2 R^2/\alpha$. The solutions in this case are known. Outside the finite potential well, the wave function $u(r)$ drops off exponentially, i.e. $\phi(r) = u(r)/r \propto \exp(-\kappa r)/r$, on the length scale

$$\frac{1}{\kappa} \sim \frac{1}{\sqrt{V_0 - E}} = \frac{R}{\sqrt{\alpha(1-\gamma)}} > \frac{1}{m}. \tag{42}$$

We see that we cannot localize the $\phi$-profile in a region smaller than $1/m$ without an instability.

Finally, we note that higher interaction terms might be introduced to cure the tachyonic potential problem above. An example would be a positive $\lambda\phi^4$ term. If this term is large enough to cure a possible instability, we would also expect its repulsive effect to make the bound state larger. Thus, we expect that our conjecture can not be avoided by extending the model in this way. However, more complex models of this type clearly require a detailed analysis, which we have to leave to future research.

## 4.2 Quartic interactions

We consider the $U(1)$-invariant theory of a complex scalar field with quartic potential,

$$V(\phi) = \frac{m^2}{2}|\phi|^2 + \lambda|\phi|^4, \tag{43}$$

taking the interaction to be attractive, $\lambda < 0$. We want to think of this as an effective theory below some cutoff $\Lambda$. This cutoff may be high enough to allow for $|\phi|$-values for which $V(\phi) < 0$. In this case the vacuum at $\phi = 0$ is only metastable.[16] In Sect. 4.2.4, we will introduce a $|\phi|^6$-term that bounds the potential from below and restricts the validity of (43) to the region below some maximal $|\phi|$-value. One may view this as an explicit implementation of a cutoff.

---

[16] The lifetime of the $|\phi| = 0$ vacuum has an exponential $1/\lambda$-dependence and a power-like dependence on the cutoff $\Lambda$. The latter comes from the running of $\lambda$ and from the prefactor of the exponential tunneling suppression [62]. We accept that the lifetimes of the vacuum and of bound states may be only finite.

In the above setting, one could in principle imagine bound states to occur based on the attractive scalar interaction alone, on gravity, or on a combination of both. We will start our discussion with the gravitationally coupled case. It is clear that for sufficiently dilute systems gravity dominates and stable boson stars exist. Moving from there into a regime of high density and small radius, where the $\phi^4$-attraction becomes important, one might naively expect that much denser objects can form. Instead, an instability develops. This suggests that small bound states, which could violate our conjecture, are excluded. In the case without gravity, we argue that bound states do not exist at all. Finally, we will allow for higher-order self-interactions, removing the instability at large $|\phi|$. Now bound states in the form of Q-balls are possible. However, they are always large enough to respect our conjecture, at least in the regime where they can be created adiabatically.

### 4.2.1 Gravitationally bound states

At small particle numbers and correspondingly large bound-state radii $R \gg 1/m$, we expect the total attractive force to be dominated by gravity. To confirm this, we estimate the relative importance of the binding energy from self-interactions. We introduce the conserved particle number (17) and, using the mean field approach with $\phi(t, \mathbf{x}) = e^{-i\omega t}\phi(\mathbf{x})$, express it in terms of the average field excursion $\bar{\phi}^2 = \int \mathrm{d}^3 x \, |\phi(\mathbf{x})|^2 / R^3$ of a localized solution:

$$N \sim m\bar{\phi}^2 R^3. \tag{44}$$

Here we also used that we are in the non-relativistic regime, $\omega \sim m$. The binding energy from self-interactions may then be expressed in terms of the boson star mass $M \sim mN$:

$$E_{\text{self-int}} \sim -\int \mathrm{d}^3 x \, |\lambda| |\phi|^4 \sim -|\lambda| R^3 \bar{\phi}^4 \sim -\frac{|\lambda| N^2}{m^2 R^3} \sim -\frac{|\lambda| M^2}{m^4 R^3}. \tag{45}$$

It is subdominant to both the gravitational binding energy $\propto 1/R$ and the quantum pressure $\propto 1/R^2$, cf. (12). Hence the radius depends on the mass just as in the familiar free-boson case, cf. $R(M)$ of (13).

As we increase the particle number and mass of the boson star, its size decreases and the attractive self-interaction becomes more relevant. Such a gravitationally bound system with additional self-interaction has been studied numerically in [30,37] (along with some analytical estimates) using the Gross-Pitaevskii-Poisson equations. As expected, the stable radius of a bound state is, for small particle numbers, the same as in (13). The additional attractive force only affects the $\mathcal{O}(1)$-prefactor:

$$R_{\text{G+SI}}(M) \sim R_{\text{FB}}(M) \sim \frac{1}{M}\left(\frac{M_{\text{P}}}{m}\right)^2. \tag{46}$$

We sketch the two curves $R_{\text{G+SI}}(M)$ and $R_{\text{FB}}(M)$ in Fig. 4.

As is also sketched in Fig. 4, there exists a maximum mass at which the curve $R_{\text{G+SI}}(M)$ ends [30,37],

$$M_{\text{G+SI}}^{\text{max}} \sim \frac{M_{\text{P}}}{\sqrt{|\lambda|}}. \tag{47}$$

Above this mass there is no regular solution of the equations of motion that can be found using the numerical approach. Depending on the coupling strength $|\lambda|$, we distinguish two scenarios: For small couplings $|\lambda| \lesssim m^2/M_{\text{P}}^2$, self-interactions remain weak and the relation (46) is valid all the way to the mass at which the star collapses to a black hole of size $R_{\text{BH}} \sim 1/m$. For stronger coupling, $|\lambda| \gtrsim m^2/M_{\text{P}}^2$, the solution breaks down before the critical radius of gravitational collapse is reached,

$$R_{\text{G+SI}}(M_{\text{G+SI}}^{\text{max}}) \sim \sqrt{|\lambda|}\frac{M_{\text{P}}}{m^2} \gtrsim \frac{1}{m}. \tag{48}$$

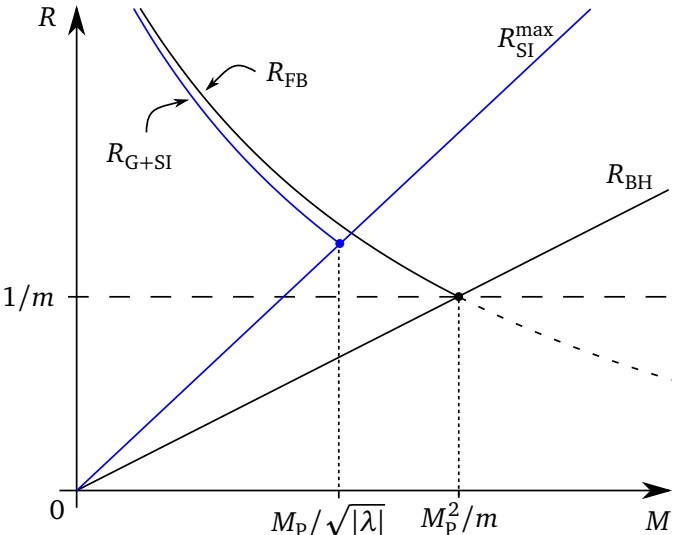

Figure 4: The $M$–$R$-plots of gravitationally bound boson stars. For the free-boson case, the radius is denoted as $R_{\text{FB}}(M)$, for the case with additional self-interactions as $R_{\text{G+SI}}(M)$. Also shown are the black-hole radius $R_{\text{BH}}(M)$ as well as the radius $R_{\text{SI}}^{\max}(M)$ at which self-interactions start to dominate the total energy, see subsection 4.2.2.

We will analyze in a moment what happens at this point. Throughout the rest of this section, our focus will be on the interesting case of large coupling (or weak gravity) $|\lambda| \gtrsim m^2/M_{\text{P}}^2$.

Note that, according to the quoted numerical results, the curve $R_{\text{G+SI}}(M)$ in the plot never falls *parametrically* below $R_{\text{FB}}(M)$ in the stable bound-state regime. Hence, $R \gtrsim 1/m$ and we are justified in the use of $\omega \sim m$ for all bound states discussed so far.

### 4.2.2 Instability from self-interactions

The total energy of a bound state of non-relativistic constituents is the sum of the localization energy

$$E_{\text{loc}} \sim \int \mathrm{d}^3x \, |\nabla\phi|^2 \sim R\bar{\phi}^2 \sim \frac{N}{mR^2}, \tag{49}$$

the binding energy from self-interactions, cf. (45),

$$E_{\text{self-int}} \sim -\int \mathrm{d}^3x \, |\lambda| |\phi|^4 \sim -|\lambda| R^3 \bar{\phi}^4 \sim -\frac{|\lambda| N^2}{m^2 R^3}, \tag{50}$$

and the gravitational energy

$$E_{\text{grav}} \sim -\frac{m^2 N^2}{M_{\text{P}}^2 R}. \tag{51}$$

Here, we used (44). By the same arguments as in Sect. 4.1 below (35), the energy associated with the mass term in the potential, $E_{\text{mass}} \sim Nm^2/\omega$, is not relevant. We sketch the total energy in Fig. 5.

For small $N$, we find a non-trivial minimum arising from the interplay of the gravitational energy and the localization energy. The resulting stable radius, as explained above, is (46). At smaller radii, there is a maximum of the total energy at

$$R_{\text{SI}}^{\max}(N) \sim \frac{N|\lambda|}{m} \sim \frac{M|\lambda|}{m^2}. \tag{52}$$

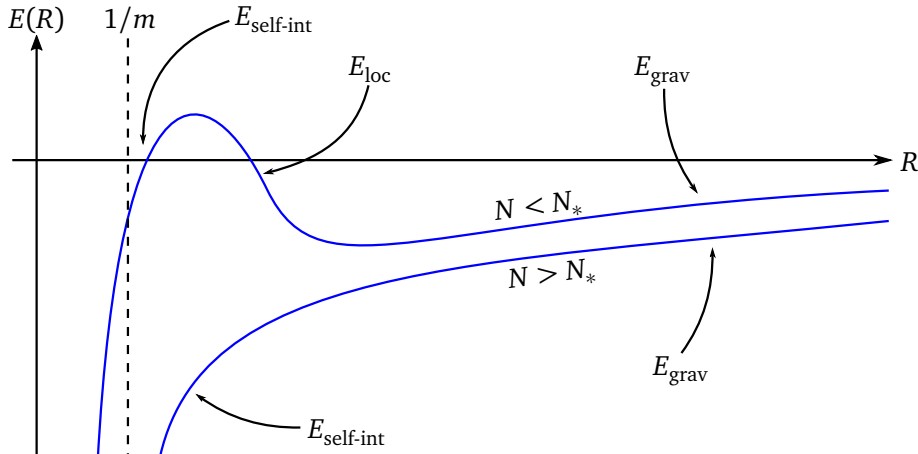

Figure 5: The total energy $E_{\text{tot}} = E_{\text{loc}} + E_{\text{grav}} + E_{\text{self-int}}$ as a function of $R$ within the non-relativistic approximation. There is a local minimum for small $N$, which disappears at large $N$. While the minimum is always to the right of the vertical line $R \sim 1/m$, the maximum can be on either side. For sufficiently large particle number, $N \gtrsim 1/|\lambda|$, the latter is also to the right of $1/m$, as for the upper curve in the sketch.

We can see that there exists a particle number $N_*$ at which the local minimum disappears. This happens at

$$R_{\text{SI}}^{\text{max}}(N_*) \sim R_{\text{G+SI}}(N_*) \qquad \Leftrightarrow \qquad N_* \sim \frac{M_{\text{P}}}{\sqrt{|\lambda|}\,m}, \tag{53}$$

in agreement with the maximum particle number or mass (47) found numerically, $N^{\text{max}} \sim N_*$. Above this particle number, the stabilizing effect of quantum pressure is not strong enough to overcome the attractive self-interaction, cf. Fig. 5. The intersection of the curves $R_{\text{SI}}^{\text{max}}$ and $R_{\text{G+SI}}$ can also be seen in Fig. 4.[17]

We should consider what happens to a bound state at the maximum particle number $N^{\text{max}}$ where the energy minimum disappears. If we keep adding particles, the configuration might collapse to a small black hole of particle number $N \sim N^{\text{max}}$ and radius $R_{\text{BH}}(M \sim M^{\text{max}}) \lesssim 1/m$. However, assuming that in the collapse process the mean-field approximation still holds, one can check that the average field value $\bar{\phi}$ inside the configuration at fixed mass $M \sim M^{\text{max}}$ and particle number $N \sim N^{\text{max}}$ necessarily exceeds the value $m^2/|\lambda|$.[18] That is, the true vacuum at $|\phi| \neq 0$ becomes classically accessible and, as long as there are no stabilizing higher-order terms in the potential, we expect vacuum decay to set in. In this way, our conjecture would not be violated. Clearly, we can not be certain of the validity of this mean-field logic during the potentially violent collapse process. More scrutiny is needed to establish the result of the collapse.

### 4.2.3 Comments on the non-gravitational theory

Finally, we want to consider a bound state from self-interaction alone, $M_{\text{P}} \to \infty$. Instead of gravitationally building up a large bound state until self-interaction takes over, we want to consider the self-bound few-particle case.

The energy barrier that can be anticipated from the many-particle mean field calculation is sketched in Fig. 5 (without the gravitational tail at large $R$). It can also be derived in the

---

[17] Note that the curve $R_{\text{SI}}^{\text{max}}(N)$ or $R_{\text{SI}}^{\text{max}}(M)$ characterizes the position of the maximum of $E_{\text{loc}}(R) + E_{\text{self-int}}(R)$. As such, it does not describe a stable state and its only meaning is to specify the point where the curve $R_{\text{G+SI}}(M)$ ends.

[18] The relevant equation is $M \sim \int \mathrm{d}^3 x \left( |\dot{\phi}|^2 + |\nabla\phi|^2 + V(\phi) \right) \sim N^2/(\bar{\phi}^2 R^3) + R\bar{\phi}^2 + R^3 V(\bar{\phi})$. Since the collapse sets in within the non-relativistic regime, we can fix the mass at $M = mN$.

few-particle scenario: Calculating the energy of a two-particle state with Hamiltonian $H$ corresponding to (43), where the particles are Gaussian wave packets of width $R$, one finds [19]

$$E_{\text{tot}}(R) = \langle H \rangle_{\text{2-particle}} \sim \begin{cases} m + \frac{1}{mR^2} - |\lambda| \frac{1}{m^2 R^3} & R \gg \frac{1}{m}, \\ \frac{1}{R} - |\lambda| \frac{1}{R} & R \ll \frac{1}{m}. \end{cases} \tag{54}$$

The non-relativistic expression contains the mass contribution $m$, the kinetic term $p^2/m \sim 1/(mR^2)$, and a contribution $\propto 1/R^3$ associated with the scalar attraction.[20] The relativistic result can only depend on the scale $1/R$, as the mass $m$ becomes irrelevant. This is sufficient to explain the second line of (54).

Let us start our analysis in the perturbative regime, $|\lambda| \lesssim 1$, and at large $R$. Here, the first line of (54) is applicable and the repulsive quantum pressure dominates. When moving to smaller $R$, this remains true until, at $R \sim 1/m$, we reach the applicability range of the second line of (54). But here, again, repulsion dominates. Thus, we have repulsion for all $R$ and no binding is possible.[21]

For strong coupling, $|\lambda| \gtrsim 1$, tunneling to large $\phi$-values becomes fast since the $\exp(-1/\lambda)$-suppression is ineffective. Hence we are forced to set a low cutoff $\Lambda \lesssim m/\sqrt{|\lambda|} \ll m$ to avoid this fast instability. Bound states exist, but they can as a matter of principle not be smaller than $1/\Lambda$, which is above the scale $1/m$.

Finally, we return to $|\lambda| \lesssim 1$ but allow for $N$ particles. We expect the interaction energy to scale as $N^2$ (every particle interacts with every other particle) and the kinetic energy to scale as $N$. For sufficiently small $N$, the two-particle discussion above will suffice. However, for $N \gtrsim 1/|\lambda|$, the energy maximum of the non-relativistic regime comes to lie at $R > 1/m$ and hence becomes trustworthy. At smaller values of $R$, left of this maximum, the energy falls first as $1/mR^2$ and then as $1/R$, in the relativistic regime. So at best we can hope for a singular bound state, with the same problem of vacuum decay as above.

### 4.2.4 Higher-order self-interactions and Q-balls

We now want to explore the limitations of our conjecture: Does it still hold if we allow for higher-order, non-renormalizable terms in the effective Lagrangian? As we argued above, gravitational bound states may be driven to instability, where $|\phi|$ grows and explores the potential in the regime dominated by the negative $\lambda|\phi|^4$ term. This instability may be cured by higher-order terms, e.g. a repulsive $|\phi|^6$ interaction. In this case, a different, not yet discussed form of a stable or metastable bound state may arise. The key feature of the potential on which this relies is the second minimum at $|\phi| \neq 0$.

Let us start with the simple example

$$V(\phi) = \frac{1}{2} m^2 |\phi|^2 \left( 1 - \frac{|\phi|^2}{\phi_0^2} \right)^2, \tag{55}$$

which corresponds to the potential in (43), if $\lambda = -m^2/\phi_0^2$, with an additional $|\phi|^6$-term. The coefficient of the latter is adjusted to ensure that the second minimum at $|\phi| = \phi_0$ is degenerate

---

[19] We use the state $|2\rangle = \int \mathrm{d}^3 p\, \mathrm{d}^3 q\, f_R(\mathbf{p}) f_R(\mathbf{q}) |\mathbf{p}, \mathbf{q}\rangle$ with $f_R(\mathbf{p}) = \mathcal{N} \exp\left(-\frac{1}{2} \mathbf{p}^2 R^2\right)$ with some normalization $\mathcal{N}$. The Fourier transform gives spatial Gaussian wave packets of width $R$.

[20] For a quantum mechanical derivation see [63]: A well-known result is that the 2-particle interaction induced by a $\phi^4$-term is described in the Schrödinger equation by a potential $V(\mathbf{x}) = -|\lambda|/m^2 \delta^{(3)}(\mathbf{x})$. Smearing out the delta-potential over a region $R^3$ results in an energy eigenvalue $E_\delta \sim -|\lambda|/(m^2 R^3)$. This can be understood since $|\lambda|/m^2$ comes from $V(\mathbf{x})$ and, with it, $1/R^3$ on dimensional grounds.

[21] There have been some attempts at constructing two-particle bound states in $\phi^4$-theory using different techniques, see [64–68] for an incomplete list. Conversely, it has also been claimed that such bound states do not exist [69]. Our simple scaling argument supports the latter option.

with the minimum at $\phi = 0$.[22] Models of this type are well-studied and their bound states are known as $Q$-balls [70] or, more generally, non-topological solitons, see e.g. [61, 71, 72].

The existence of bound states is easily understood analytically within a thin-wall approximation: In the inner region of the $Q$-ball of radius $R$ one has $\phi = \phi_0 \exp(-i\omega t)$. This region is surrounded by a wall of thickness $D \ll R$, in which the field transits from the $|\phi| = \phi_0$ to the $\phi = 0$ minimum. It is easy to show that $D \sim 1/m$ and that the wall tension is $T \sim D m^4/|\lambda|$. The expression $N \sim \omega \phi_0^2 R^3$ for the particle number, together with $\lambda = -m^2/\phi_0^2$, can be used to express $\omega$ in terms of $N$, $\lambda$, and $R$. With this, one can write the total energy, which comes from the inner region and the bubble wall, as a function of $R$:

$$E = E_{\text{inner}} + E_{\text{wall}} \sim \omega^2 \phi_0^2 R^3 + T R^2 \sim \frac{N^2 |\lambda|}{m^2 R^3} + \frac{m^3 R^2}{|\lambda|}. \tag{56}$$

The $Q$-ball radius follows by minimization,

$$R \sim \frac{(|\lambda| N)^{2/5}}{m}, \qquad \text{such that} \qquad E \sim \frac{mN}{(|\lambda| N)^{1/5}}. \tag{57}$$

Since the $Q$-ball can decay to $N$ free particles if $E \geq mN$, we must have $N \gtrsim N_c \sim 1/|\lambda|$ as a stability requirement. But this implies that $R \gtrsim 1/m$, so our conjecture can not be violated in the thin-wall approximation. If we modify the potential such that the $\phi_0$-minimum has positive vacuum energy, the decay is facilitated and $N_c$ increases. This leads to an increase of $R$ and hence our conjecture is even more safe.

We also note that, according to (53), $Q$-balls of this type can be adiabatically produced by driving boson stars beyond the boundary of stability: Indeed, since $M_P/m > 1/\sqrt{|\lambda|}$ in the region of interest, we always have $N > N_c$ and the resulting $Q$-balls are stable.

We now turn to the possibly more critical case in which the vacuum energy at the second minimum is negative:

$$V(\phi) = \frac{1}{2} m^2 |\phi|^2 - |\lambda| |\phi|^4 + \frac{1}{\Lambda^2} |\phi|^6, \qquad \text{with} \qquad \Lambda^2 > \frac{2m^2}{\lambda^2}. \tag{58}$$

While the $\phi = 0$ vacuum and hence any possible $Q$-balls are now at best metastable, they can certainly be very long-lived. $Q$-balls relying on such a negative-energy second minimum are frequently called $Q$-bubbles. As long as the depth of the second minimum is parametrically small compared to the barrier height, the thin-wall approximation is useful and an analytical treatment is possible [73]. Unsurprisingly, our previous discussion of the case with degenerate minima still applies and the bound state conjecture can not be violated. However, it also follows from the discussion above that bound states which are smaller by $\mathcal{O}(1)$ factors are now conceivable.

Thus, the critical question is whether by going to the regime of a *deep* second minimum one can find parametrically small $Q$-bubbles, violating the conjectured bound $1/m$. Unfortunately, we were not able to extract an unambiguous answer from the partial analytical results of [74]. Also from the numerical results of [75, 76] the answer is at least not obvious: The authors use the dimensionless quantities $\widetilde{\omega} = \omega/(|\lambda| \Lambda)$ and $\widetilde{m} = m/(|\lambda| \Lambda)$ as well as an analogously rescaled particle number or charge $\widetilde{N} = |\lambda| N$. Making use of the fact that a parametrically deep second minimum is located at $\phi_0 \sim \sqrt{|\lambda|} \Lambda$, one can estimate the radius in terms of these parameters as follows:

$$R \sim \left( \frac{N}{\phi_0^2 \omega} \right)^{1/3} \sim \frac{1}{m} \left( \frac{\widetilde{m}^3 \widetilde{N}}{\widetilde{\omega}} \right)^{1/3}. \tag{59}$$

---

[22] Writing the $|\phi|^6$-term as $|\phi|^6/\Lambda^2$, we find that $\Lambda^2 = 2\phi_0^2/|\lambda|$, thus fixing the cutoff of the model.

Inserting concrete numerical values for $\widetilde{\omega}$, $\widetilde{m}$ and $\widetilde{N}$ from [75], we only arrive once again at $\mathcal{O}(1)$ coefficients multiplying $1/m$. Presumably, a dedicated numerical study would be needed to settle our question about small $Q$-bubbles.

We note once again that, even if small $Q$-bubbles exist, creating them adiabatically is problematic since boson stars collapse only at $N \gtrsim 1/|\lambda|$. Hence small $Q$-bubbles are presumably out of reach in the model discussed above. However, once we allow for a $|\phi|^6$-term, nothing stops us from also adding $|\phi|^8$- or $|\phi|^{10}$-terms etc. Then one may as well 'draw scalar potentials by hand'. In such a general setting, it is easy to imagine that $Q$-balls of size $\ll 1/m$ may, after all, both exist and be constructed adiabatically.

Thus, we prefer not to claim that parametrically small $Q$-balls do not exist or can not be built. Instead we have, partially with a view on this subsection, formulated our conjecture as a statement about power-counting renormalizable effective field theories. A possible extension beyond this set of models is left to future research.[23]

Finally, another possible concern is the existence of $Q$-balls in renormalizable models [78]. This requires more fields and an analysis of our conjecture in this context goes beyond the scope of the present paper.[24] It would however be important to understand whether such $Q$-balls can be small compared to all mass scales governing the theory in the original vacuum and whether they can be constructed adiabatically. A positive answer may force us to search for stronger constraints than renormalizability.

## 4.3 Axions

We now turn to the example of axion bound states. The relevant potential is

$$V(\phi) = m^2 f^2 [1 - \cos(\phi/f)] = \frac{m^2}{2}\phi^2 - \frac{m^2}{f^2}\phi^4 + \mathcal{O}\big((\phi/f)^6\big). \tag{60}$$

As $\phi$ is real, we are lacking the notion of the exactly conserved particle number used in Sects. 4.1 and 4.2. Nevertheless, at low energies approximate particle number conservation holds and long-lived axion stars (or, more generally 'oscillatons') exist. Moreover, detailed numerical simulations of the coupled Klein-Gordon and Einstein equations are available. In the following, we will consider these simulations and ask for small bound states, potentially violating our conjecture. For recent work on axion stars see e.g. [80, 81] and refs. therein.

Specifically, we will rely on the 'phase diagram' obtained in [41, 44] and sketched in Fig. 6. It includes the curve $f_{\min}(M)$, representing the minimum axion decay constant for which an axion star with mass $M$ is stable. Note that it is customary to use the dimensionless variable $Mm/M_P^2$ to characterize the mass of the star. In our context, it may be more intuitive to move through this plot horizontally, at fixed $f$. Then the curve $f_{\min}$ specifies the maximum mass up to which the star remains stable. We will discuss the different regions of the diagram in turn.

We start with region I, containing the free limit $f \to \infty$. Here, $f_{\min}(M)$ approaches the vertical line $Mm/M_P^2 = \mathcal{O}(1)$, where a free boson star would collapse to a black hole. As long as $f/M_P \gtrsim 1$, the curve $f_{\min}$ deviates from the free-boson vertical line only by an $\mathcal{O}(1)$ factor. Thus, no parametrically small black holes can form and our conjecture is safe.

---

[23] Let us note that a different idea for probing UV physics with $Q$-balls appeared in [77]. It does not aim at small bound-state radii but rather employs large VEVs to catalyze certain UV-scale-suppressed transitions between light particles.

[24] An analysis based on the toy model $V(\phi) = m^2|\phi|^2 - A|\phi|^3 + \lambda|\phi|^4$ mimicking renormalizable couplings between multiple fields suggests the following: A thin-wall calculation for degenerate vacua gives a minimal charge $N$ for stable $Q$-balls that leads to $R \gtrsim 1/m$, similar to the discussion following (57). The thick-wall limit at small charges and very non-degenerate vacua gives $R \sim 1/(\epsilon m)$ with $\epsilon \ll 1$ [79]. Of course, a model with multiple fields still requires a proper, independent analysis.

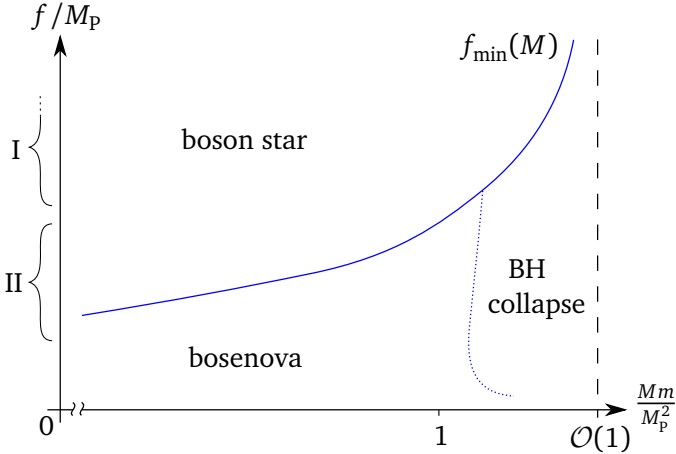

Figure 6: Sketch of the axion star phase diagram of [41, 44]. Region I contains the free limit, $f/M_P \gg 1$. In the intermediate region II, the theory is approximated by an attractive $\phi^4$-theory in the dilute regime. Below region II, corresponding to large $\phi/f$, higher-order terms of the potential (60) are relevant. The dashed vertical line characterizes the mass at which a free boson star collapses to a black hole.

In region II, the graph is approximately linear:

$$\frac{f_{\min}(M)}{M_P} \sim \frac{mM}{M_P^2}. \tag{61}$$

One may worry that a star which becomes unstable by crossing this line collapses into a small black hole. After all, we are now far below the critical value $M \sim M_P^2/m$ distinguished by our conjecture. However, as already indicated in Fig. 6, this is not what happens [41, 44]: The instability manifests itself through the emission of relativistic axions in form of a 'bosenova' [44, 82]. No small black hole is formed.

In more detail, the fate of axion stars becoming unstable in region II is as follows [38]: After shedding an outer shell in a bosenova, a dense axion star remnant forms. It is stabilized by higher-order (in the expansion of the cosine potential) repulsive interactions. This new dense object is then stable up to a total mass $\sim 10^5 M_{FB}^{\max}$, for scalar masses $m \sim 10^{-4}\,\mathrm{eV}$ (typical of a QCD axion). So instead of creating a small black hole violating our bound, the object goes through an unstable transition to a different stable configuration, where it may reach larger masses before collapsing to a black hole, cf. Fig. 7. The dense branch satisfies (15) for all masses. Similar results were reported in [83] and very recently additional dense branches have been found in [84]. The radii on these branches and at small $f$ can be even closer to the black hole radius. Analytical approximations were provided in [85]. See also [42] for a discussion on the stability and lifetime of the axion star in the dense regime.

It would be interesting to include an analysis of the size of non-gravitational bound states of real fields, i.e. oscillons [86]. We were not able to extract enough information from the largely numerical studies to repeat our semi-quantitative axion-star discussion for the oscillon case. However, we expect the interesting regime of small oscillons to experience the same instability through violent particle production as seen in the gravitational case: According to [87], the metastability of purely field-theoretic bound states of real scalar fields is due to the approximate $U(1)$-symmetry of their effective low-energy description through a complex scalar. In other words, the finiteness of the lifetime comes from the explicit breaking of this symmetry at high energies. We expect the regime of $R \lesssim 1/m$, the relativistic regime, to break this $U(1)$-symmetry significantly. Particle production would then become efficient and prevent the existence of small, long-lived oscillons. Nevertheless, this is only an expectation and more work is needed to establish it.

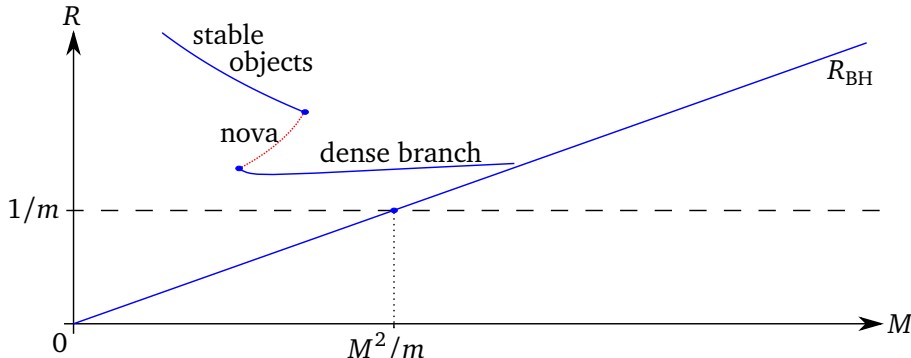

Figure 7: Sketch of an axion star's trajectory in the $M$–$R$-plane as discussed in [38]. In drawing this for general $m$, we extrapolate from the graph given for $m \sim 10^{-4}$ eV.

## 4.4 Bound states involving non-scalar particles

We have so far only discussed scalar particles bound by scalar forces. The reason is that we view such bound states as most critical in terms of providing a counterexample to our conjecture. Here is a short list of other possibilities which we consider less dangerous:

First, free, massive vectors can form non-topological solitons [88] or gravitationally bound Proca stars [36]. The parametric behavior appears to be similar to the corresponding scalar objects.

Second, when binding fermions one faces additional repulsion due to Fermi pressure. This leads to the Chandrasekhar limit $\sim M_{\rm P}^3/m^2$ for the mass of a fermion star, exceeding the critical boson star mass by $M_{\rm P}/m$. The radius exceeds $1/m$ by the same large factor.[25]

Finally, the binding may be due to a vectorial (i.e. gauge) force rather than to a scalar force. In the abelian case, one is limited to two constituents since the charges have to be opposite. The size is $1/(g^2 m)$, consistent with our conjecture at weak coupling.[26] At strong coupling, it is natural to use the dual, weakly coupled description. Thus, we have to discuss binding magnetic monopoles in a weakly coupled electric theory. Such monopoles are extended with a size comparable to their inverse mass, enhanced by the strong magnetic coupling. So our conjecture appears to be safe. In the non-abelian case, the confinement scale sets both the size and mass of particles and bound states. Again, we see no prospects for violating our conjecture parametrically, except maybe if one involves the rank of the gauge group as a large parameter. It could be worthwhile to further study this.

## 5 Summary and discussion

Our analysis of swampland ideas for constraining scalar interactions has lead us to a novel proposal: the Bound State Conjecture (15). It differs from the conventional swampland framework in that it remains non-trivial when gravity is completely decoupled. Hence, it may turn out to be a provable feature of a class of non-gravitational QFTs.

We have described previous approaches in Sect. 2. Their main idea is to construct an inequality that quantifies the statement 'gravity is the weakest force' in the presence of scalar interactions. We have also presented problems, or possible counterexamples, we see with these proposals. Our approach, as discussed in Sect. 3, takes a different route: We do not attempt to forbid bound states by requiring that repulsive forces outweigh gravitational attraction.

---

[25] Mixed fermion-boson stars have been studied (in gravitationally bound systems) in [89] and more recently in e.g. [90].

[26] One can compare this to the size of a bound state bound by a scalar mediator (20) extrapolated to small $N$.

Neither do we try to claim that scalar attractive forces must act more strongly than gravity. Instead, our premise is that bound states should not be forbidden but constrained. Specifically, there should be a minimal size for bound states. We have quantified this by stating that the smallest black hole that can be built adiabatically from individual particles in an interacting theory is not parametrically smaller than the one built from free scalar particles. This does not give rise to a 'weak gravity' conjecture. Rather, it claims that 'attractive forces can not be parametrically stronger than gravity alone'. By calculating the size of a black hole that can be adiabatically built from free scalars, one finds that this statement is actually (maybe surprisingly) independent of the strength of gravity: The resulting radius and hence the minimal size is $R \gtrsim 1/m$, with $m$ the mass of the free scalar. We stress that this result is independent of $M_{\mathrm{P}}$. We have put the resulting 'Bound State Conjecture' to the test in Sect. 4. In all examples it has turned out that either the model becomes tachyonic or a bosenova-type instability of the bound state develops if one tries to beat the conjectured minimal radius $R \sim 1/m$.

We conclude with a short list of open problems and comments:

An obvious open problem is to determine a general, purely field-theoretic origin of the bound. The non-gravitational formulation should, if correct, allow for a proof using the well-understood framework of QFT. Employing the uncertainty relation $p \sim 1/R$, in essence, this would involve proving that relativistic particles can not be bound. Another way forward might be to consider causality constraints in scattering processes involving the bound states, as it was done in [56, 57]. In this context we should also note that the decoupling of gravity remains peculiar. That is, we do not see any a priori reason why $M_{\mathrm{P}}$ should drop out of the bound. In other words, why do the two limitations for small bound states, one from field-theoretic instabilities and one from horizon formation, both coincide and give the value $R \sim 1/m$?

Next, we want to highlight that the critical radius $R \sim 1/m$ is independent of the space-time dimension $d$: To see this, one repeats the original derivation, where we asked for the radius at which the localization energy $E_{\mathrm{loc}} \sim (M/m)/(mR^2)$ and the gravitational energy $E_{\mathrm{grav}} \sim M^2/(M_{\mathrm{P}}^{d-2}R^{d-3})$ coincide. Then, requiring that this radius is also the black-hole radius corresponding to $M$, one arrives at $R \sim 1/m$ for general $d$. However, this result is only formal since in $d \geq 6$ dimensions, the energy $E_{\mathrm{loc}} + E_{\mathrm{grav}}$ has a maximum rather than a minimum. Thus, particles cannot be bound by gravity at large distances at all. The case $d = 5$ is special since $E_{\mathrm{loc}}$ and $E_{\mathrm{grav}}$ scale identically with $R$. There is then a critical particle number $N_c \sim (M_{\mathrm{P}}/m)^3$ for which the particles can be brought close together at no energy cost to form a black hole of size $1/m$.

Furthermore, we should warn the reader that, when moving outside the domain of power-counting renormalizable theories, small bound states appear conceivable. This is suggested by our discussion of small 'Q-bubbles' relying on a deep true vacuum in Sect. 4.2.4. Of course, the price to pay is that now our basic vacuum is a false one, being hence only long-lived rather than stable. We must also admit that we have neither established that such spiky Q-bubbles really exist nor do we see how to construct them adiabatically. The problem of small bound states in more involved multi-field models has also not yet been studied by us. These two open questions appear to be the most promising routes to either disprove the conjecture, to limit its generality, or, on the contrary, to collect highly non-trivial evidence in its favor.

Finally, let us point out a possible version of our conjecture related to resource theory. In (quantum) resource theory one defines so-called free states, which are readily available, and free operations, which the experimenter can perform. In our context, these would be light particles (of mass $\sim m$) and adiabatic processes involving them. The resources are then states that have special value since they can not be produced from the above. These, in our case, would be small bound states or, possibly equivalently, fundamental heavy particles with mass $M \gg m$. The latter would become available through unsuppressed transition amplitudes from sufficiently heavy and localized bound states. The special feature setting these resource states

apart may be some kind of strong entanglement involving constituent particles at relativistic momenta. It would be interesting to establish a formulation of our conjecture in these terms more carefully. The conjecture might then represent an obstacle to building an 'IR-to-UV transformer', i.e. a device that is fed light particles and which produces a heavy, UV-scale, fundamental particle after enough energy has been supplied in this IR channel.

## Acknowledgements

We thank Diego Hofman, Joerg Jaeckel, Sandra Klevansky, Florent Michel and Christian Reichelt for discussions. This collaboration was in part initiated at the Aspen Center for Physics, which is supported by National Science Foundation grant PHY-1607611. This work is supported by the Deutsche Forschungsgemeinschaft (DFG, German Research Foundation) under Germany's Excellence Strategy EXC 2181/1 - 390900948 (the Heidelberg STRUCTURES Excellence Cluster) and the Graduiertenkolleg 'Particle physics beyond the Standard Model' (GRK 1940). BF is supported in part by ERC Consolidator Grant QUANTIVIOL.

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
