# Peer review of "A Conjecture on the Minimal Size of Bound States"

_SciPost Physics, doi:SciPost Phys. 8, 058 (2020)_

## Round 2 · Referee Report · Anonymous · 2020-3-6

Strengths

1) Counterexample to a conjecture that exists in the literature.
2) Reasonable new conjecture offered with good amount of supporting evidence

Weaknesses

1) Some points unclear (see below)

Report

This is a nice paper with a novel, reasonable Swampland conjecture. In general, there are too many Swampland conjectures in our field, but this paper does the field a service by first providing a counterexample to another such conjecture, so I think they are justified in making a new one.

I have an issue with the phrasing of the “Bound State Conjecture” on page 9 and the discussion immediately surrounding it. I had to read section 3.2 and understand Figure 1 before I could make sense of their Bound State Conjecture: the key point is that by an “adiabatic” construction, they want to exclude the possibility of a smaller black hole arising by Hawking evaporation: such a Hawking evaporation process is not “adiabatic,” so it does not violate the assumptions of the conjecture. Here’s my point: the authors ought to explicitly spell this out immediately after they state the conjecture on page 9: “Note that this leaves open the possibility of a smaller black hole arising via a Hawking evaporation process, as such a process is not ‘adiabatic.’” Or something like that.

There is something funny about their non-gravitational conjecture on page 12: the conjecture holds that there cannot exist a bound state with radius smaller than 1/m, with m the most massive stable particle in the theory. On the other hand, they define any state with mass m_0 and radius smaller than 1/m_0 to be a particle, as opposed to a bound state. So, if m_0 is smaller than m, then a radius r smaller than 1/m would also be smaller than 1/m_0, so we’d call it a particle, not a bound state. Only if m_0 is larger than m do we have a nontrivial bound: namely, r cannot be smaller than 1/m. Yet if r is smaller than 1/m_0, we’d call this a particle, not a bound state. So really the conjecture amounts to the statement that the radius satisfies r < 1/m_0 or r > 1/m. It would be nice if this point were stated more clearly.

p. 13—“Such objects can form bound states and collapse to black holes”—I thought we were now working in an EFT context (gravity decoupled); why are we still talking about black holes?

Requested changes

1) Add discussion clarifying importance of adiabaticity regarding Hawking evaporation.
2) Point out more clearly subtlety regarding non-gravitational bound state conjecture discussed above.
3) Address point about black holes being discussed in non-gravitational context.

  • validity: high
  • significance: high
  • originality: top
  • clarity: good
  • formatting: excellent
  • grammar: perfect

Author Sascha Leonhardt on 2020-03-17
(in reply to Report 1 on 2020-03-06)
Category:
answer to question
reply to objection

We would like to thank the referee for insightful comments that will improve the clarity of the paper. Specifically, referring to the suggested changes:

(1) We agree with the referee that we should clarify immediately after the conjecture on p. 9 that a small black hole can arise by Hawking evaporation. As the referee points out, we clarify this in section 3.2, but we will already add a sentence immediately after the conjecture clarifying this.

(2) Indeed, it is correct that this bound is nontrivial only when the mass m_0 of the bound state is larger than the mass m of the heaviest stable particle. The case where m_0 > m and r < m_0 is excluded by the assumptions of our conjecture, which state that the particle of mass m is the heaviest stable particle of our theory.

We will be happy to clarify this in the discussion of the conjecture in Sect. 3.3.

(3) The referee is correct in that the mentioning of black hole collapse near the top of page 13 is neither appropriate nor necessary. The relevant sentence should be replaced as follows:

"Such objects can form bound states and collapse to black holes, so it would be wrong to demand that our conjecture holds with m = 0."
-->>
"Given some additional gauge or scalar force, such objects can form bound states of non-zero radius. So it would be wrong to demand that our conjecture holds with m = 0."

---

## Round 3 · List of Changes

- Added a comment on Hawking evaporation already in the paragraph underneath the conjecture in Sect. 3.1
- Shortened this comment underneath (13) in Sect. 3.2 correspondingly
- Added a short paragraph on the case of a bound state of mass M fulfilling R < 1/M in Sect. 3.3
- Rephrased a comment on bound states in Yang-Mills theories in the next paragraph to not include gravity

You are currently on this page

Resubmission 1912.09485v3 on 23 March 2020

---

## Editorial Decision

published